# A quantitative model used to compare within-host SARS-CoV-2, MERS-CoV, and SARS-CoV dynamics provides insights into the pathogenesis and treatment of SARS-CoV-2

Kwang Su Kim[1☉], Keisuke Ejima[2☉], Shoya Iwanami[1], Yasuhisa Fujita[1], Hirofumi Ohashi[3], Yoshiki Koizumi[4], Yusuke Asai[4], Shinji Nakaoka[5], Koichi Watashi[3,6,7,8], Kazuyuki Aihara[9], Robin N. Thompson[10,11], Ruian Ke[12,13], Alan S. Perelson[12,13‡]*, Shingo Iwami[1,8,14,15,16‡]*

1 Department of Biology, Faculty of Sciences, Kyushu University, Fukuoka, Japan, 2 Department of Epidemiology and Biostatistics, Indiana University School of Public Health–Bloomington, Bloomington, Indiana, United States of America, 3 Department of Virology II, National Institute of Infectious Diseases, Tokyo, Japan, 4 National Center for Global Health and Medicine, Tokyo, Japan, 5 Faculty of Advanced Life Science, Hokkaido University, Sapporo, Japan, 6 Department of Applied Biological Science, Tokyo University of Science, Noda, Japan, 7 Institute for Frontier Life and Medical Sciences, Kyoto University, Kyoto, Japan, 8 JST-Mirai, Japan Science and Technology Agency, Saitama, Japan, 9 International Research Center for Neurointelligence, University of Tokyo Institutes for Advanced Study, University of Tokyo, Tokyo, Japan, 10 Mathematics Institute, University of Warwick, Coventry, United Kingdom, 11 Zeeman Institute for Systems Biology and Infectious Disease Epidemiology Research, University of Warwick, Coventry, United Kingdom, 12 New Mexico Consortium, Los Alamos, New Mexico, United States of America, 13 Theoretical Biology and Biophysics Group, Los Alamos National Laboratory, Los Alamos, New Mexico, United States of America, 14 Institute for the Advanced Study of Human Biology, Kyoto University, Kyoto, Japan, 15 NEXT-Ganken Program, Japanese Foundation for Cancer Research, Tokyo, Japan, 16 Science Groove, Fukuoka, Japan

☉ These authors contributed equally to this work.
‡ These authors are joint senior authors on this work.
* asp@lanl.gov (ASP); siwami@kyushu-u.org (SI)

**Data Availability Statement:** All relevant data are within the paper and its Supporting Information files.

## Abstract

The scientific community is focused on developing antiviral therapies to mitigate the impacts of the ongoing novel coronavirus disease 2019 (COVID-19) outbreak. This will be facilitated by improved understanding of viral dynamics within infected hosts. Here, using a mathematical model in combination with published viral load data, we compare within-host viral dynamics of SARS-CoV-2 with analogous dynamics of MERS-CoV and SARS-CoV. Our quantitative analyses using a mathematical model revealed that the within-host reproduction number at symptom onset of SARS-CoV-2 was statistically significantly larger than that of MERS-CoV and similar to that of SARS-CoV. In addition, the time from symptom onset to the viral load peak for SARS-CoV-2 infection was shorter than those of MERS-CoV and SARS-CoV. These findings suggest the difficulty of controlling SARS-CoV-2 infection by antivirals. We further used the viral dynamics model to predict the efficacy of potential antiviral drugs that have different modes of action. The efficacy was measured by the reduction in the viral load area under the curve (AUC). Our results indicate that therapies that block de novo infection or virus production are likely to be effective if and only if initiated before the viral load peak (which appears 2–3 days after symptom onset), but therapies that promote

**Funding:** This study was supported in part by Basic Science Research Program through the National Research Foundation of Korea funded by the Ministry of Education 2019R1A6A3A12031316 (to K.S.K.); Grants-in-Aid for JSPS Scientific Research (KAKENHI) Scientific Research B 17H04085 (to K. W.), 18KT0018 (to S.I.), 18H01139 (to S.I.), 16H04845 (to S.I.), Scientific Research S 15H05707 (to S.N. and K.A.), Scientific Research in Innovative Areas 20H05042 (to S.I.), 19H04839 (to S.I.), 18H05103 (to S.I.); AMED JP20dm0307009 (to K.A.); AMED CREST 19gm1310002 (to S.I.); AMED Japan Program for Infectious Diseases Research and Infrastructure, 20wm0325007h0001, 20wm0325004s0201, 20wm0325012s0301, 20wm0325015s0301 (to S. I.); AMED Research Program on HIV/AIDS 19fk0410023s0101 (to S.I.); AMED Research Program on Emerging and Re-emerging Infectious Diseases 19fk0108156h0001, 20fk0108140s0801 and 20fk0108413s0301 (to S.I.); AMED Program for Basic and Clinical Research on Hepatitis 19fk0210036j0002 (to K.W.), 19fk0210036h0502 (to S.I.); AMED Program on the Innovative Development and the Application of New Drugs for Hepatitis B 19fk0310114j0003 (to K.W.), 19fk0310101j1003 (to K.W.), 19fk0310103j0203 (to K.W.), 19fk0310114h0103 (to S.I.); Moonshot R&D Grant Number JPMJMS2021 (to K.A. and S. I.) and JPMJMS2025 (to S.I.); JST PRESTO (to S. N.); JST MIRAI (to K.W. and S.I.); The Yasuda Medical Foundation (to K.W.); Takeda Science Foundation (to K.W.); Mochida Memorial Foundation for Medical and Pharmaceutical Research (to K.W.); Mitsui Life Social Welfare Foundation (to S.I. and K.W.); Shin-Nihon of Advanced Medical Research (to S.I.); Suzuken Memorial Foundation (to S.I.); Life Science Foundation of Japan (to S.I.); SECOM Science and Technology Foundation (to S.I.); The Japan Prize Foundation (to S.I.); Fukuoka Financial Group, Inc. (to S.I.); Kyusyu Industrial Advancement Center Gapfund Program (to S.I.); Foundation for the Fusion of Science and Technology (to S.I.); a Junior Research Fellowship from Christ Church, Oxford (to R.N.T.), and a U.S. National Science Foundation grant PHY-2031756 (to A.S. P.). The funders had no role in study design, data collection and analysis, decision to publish, or preparation of the manuscript.

**Competing interests:** The authors have declared that no competing interests exist.

**Abbreviations:** AUC, area under the curve; COVID-19, coronavirus disease 2019.

cytotoxicity of infected cells are likely to have effects with less sensitivity to the timing of treatment initiation. Furthermore, combining a therapy that promotes cytotoxicity and one that blocks de novo infection or virus production synergistically reduces the AUC with early treatment. Our unique modeling approach provides insights into the pathogenesis of SARS-CoV-2 and may be useful for development of antiviral therapies.

## Introduction

The ongoing coronavirus disease 2019 (COVID-19) outbreak was first reported in Wuhan, China, in late December 2019 [1,2]. Since then, the causative agent, SARS-CoV-2 (severe acute respiratory syndrome coronavirus 2), has been transmitted elsewhere in China and to most other countries and territories around the world. The number of global confirmed cases currently stands at more than 103 million (as of 3 February 2021). Given that 40%–45% of patients are asymptomatic [3], and even symptomatic infections are underreported [4], the true number of cases is most likely much higher than this.

Antiviral drugs and vaccines are currently under development to counter this outbreak. The efficacy of these drugs can be evaluated in vitro using a cell culture system supporting SARS-CoV-2 infection [5,6] and in various animal models [7–10].

To aid the development process, characterization of the viral dynamics of SARS-CoV-2 is crucial. Several studies have reported longitudinal viral load data from symptomatic patients collected for over 20 days after symptom onset [8,11–16]. Mathematical models describing viral dynamics have been used to analyze such data [17–20]. In a recent paper [9], the pathogeneses of SARS-CoV-2, MERS-CoV, and SARS-CoV infections were compared in a nonhuman primate model. Here, we analyze and compare longitudinal viral load data of SARS-CoV-2, SARS-CoV, and MERS-CoV in humans. Further, we fit a mathematical model to the viral load data and then use the model with best-fit parameters to predict the effect of potential antiviral treatments on viral dynamics. We do not consider treatments, such as dexamethasone, aimed at reducing the inflammatory response or other downstream events that can lead to the generation of symptoms. The results of our antiviral treatment simulations provide information useful for the development of antiviral agents and treatment strategies for SARS-CoV-2, specifically addressing questions such as the best time to initiate a therapy given its mode of action. Interestingly, we find that the ideal timing varies depending on the viral–host process targeted by the antiviral drug.

## Results/Discussion

### Characterizing SARS-CoV-2, MERS-CoV, and SARS-CoV infections by analyzing viral load measurements

We analyzed longitudinal SARS-CoV-2 viral load data reported in [11–14], MERS-CoV viral load data reported in [21,22], and SARS-CoV viral load data reported in [23] using a viral dynamic model (see Methods). Further details about the data sources are described in S1 Text and summarized in S1 Table. A nonlinear mixed-effect modeling approach was employed in which we fit the model to all of the patient data simultaneously to estimate parameters (see Methods). The estimated population parameters are listed in Table 1, and estimated individual parameters for each patient are listed in S2 Table. Comparing population parameters between SARS-CoV-2 and the other 2 coronaviruses, the maximum rate constant for viral replication

**Table 1. Estimated parameters (fixed effect) for SARS-CoV-2, MERS-CoV, and SARS-CoV infection.**

| Parameter name | Symbol (unit) | SARS-CoV-2 | MERS-CoV | SARS-CoV |
|---|---|---|---|---|
| Parameters in the model | | | | |
| Maximum rate constant for viral replication | $\gamma$ (day$^{-1}$) | 4 | 1.46[#] | 4.13 |
| Rate constant for virus infection | $\beta$ ((copies/ml)$^{-1}$ day$^{-1}$) | $5.2 \times 10^{-6}$ | $1.4 \times 10^{-8}$[#] | $4.9 \times 10^{-8}$[#] |
| Death rate of infected cells | $\delta$ (day$^{-1}$)[&] | 0.93 | 0.93 | 0.93 |
| Viral load at symptom onset | $V(0)$ (copies/ml) | $6.5 \times 10^{3}$ | $6.6 \times 10^{4}$ | $3.3 \times 10^{-2}$[#] |
| Quantities derived from the parameters | | | | |
| Mean duration of virus production | $L$ (days) | 1.08 | 1.08 | 1.08 |
| Within-host reproduction number at symptom onset | $R_{S0}$ | 4.30 | 1.57[#] | 4.44 |
| Critical inhibition level | $C^*$ | 0.77 | 0.38[#] | 0.75 |
| Time from symptom onset to viral load peak | $T_{\mathrm{P}}$ (days)[$] | 2.0 | 12.2 | 7.2[$] |

[#]Statistically different from SARS-CoV-2 (the Wald test).

[&]The death rate of infected cells was assumed to be the same between the viruses in the process of model selection.

[$]$T_p$ was computed from simulation, and the difference from SARS-CoV-2 was tested by the jackknife test.

($\gamma$) of SARS-CoV-2 was significantly larger than that of MERS-CoV ($p < 2.2 \times 10^{-16}$) but similar to that of SARS-CoV. The rate constant for virus infection ($\beta$) of SARS-CoV-2 was significantly larger than that of both MERS-CoV and SARS-CoV ($p = 1.0 \times 10^{-8}$ and $p = 1.3 \times 10^{-12}$, respectively). Moreover, the viral load at symptom onset ($V(0)$) of SARS-CoV-2 was similar to that of SARS-CoV, but less than that of MERS-CoV ($p < 2.2 \times 10^{-16}$). Based on the individual parameters, the best-fit viral load curves for each patient are plotted along with the observed data in S1 Fig for SARS-CoV-2, MERS-CoV, and SARS-CoV. We further calculated and compared the following quantities, which are derived from the estimated parameters or available by running the model (Table 1): the mean duration of virus production from an infected cell ($L = 1/\delta$); the within-host reproduction number at symptom onset ($R_{S0} = \gamma/\delta$), which is the average number of newly infected cells produced by a single infected cell at symptom onset (c. f. [24]); the time from symptom onset to the viral load peak ($T_{\mathrm{p}}$); and the critical inhibition level ($C^* = 1 - 1/R_{S0}$) that needs to be reached by antivirals or vaccines to ensure that the viral infection is driven to extinction [25–27].

$R_{S0}$ of SARS-CoV-2 was statistically significantly larger than that of MERS-CoV ($p < 2.2 \times 10^{-16}$) and no different from that of SARS-CoV (Table 1). Further, according to our model, SARS-CoV-2 hit its viral load peak 2.0 days after symptom onset (i.e., $T_{\mathrm{p}}$), which is earlier than MERS-CoV and SARS-CoV, which peaked at 12.2 days and 7.2 days after symptom onset, respectively; however, the difference was statistically significant only between SARS-CoV-2 and SARS-CoV ($p = 2.24 \times 10^{-6}$; Fig 1; Table 1).

Both the larger $R_{S0}$ value of SARS-CoV-2 than that of MERS-CoV and the earlier peak in viral load for SARS-CoV-2 than the other coronaviruses suggest that the virus more effectively replicates and spreads within-host than MERS-CoV and SARS-CoV. In other words, treating SARS-CoV-2 infection may require more potent therapies and therapies given earlier than for the other coronaviruses. Further, the shorter $T_{\mathrm{p}}$ of SARS-CoV-2 suggests that treating SARS-CoV-2 infection following symptom onset is more challenging because effective antiviral treatment should be initiated before the viral peak, as we demonstrate in the next section. Given that the mean time from symptom onset to hospitalization observed in China was 4.6 days [28], symptom-based diagnosis combined with antiviral treatment might not be an effective treatment strategy if treatment needs to be given in a hospital setting. In the next section, we provide a detailed analysis of anti-SARS-CoV-2 therapy varying with the drug efficacy and timing of treatment initiation.

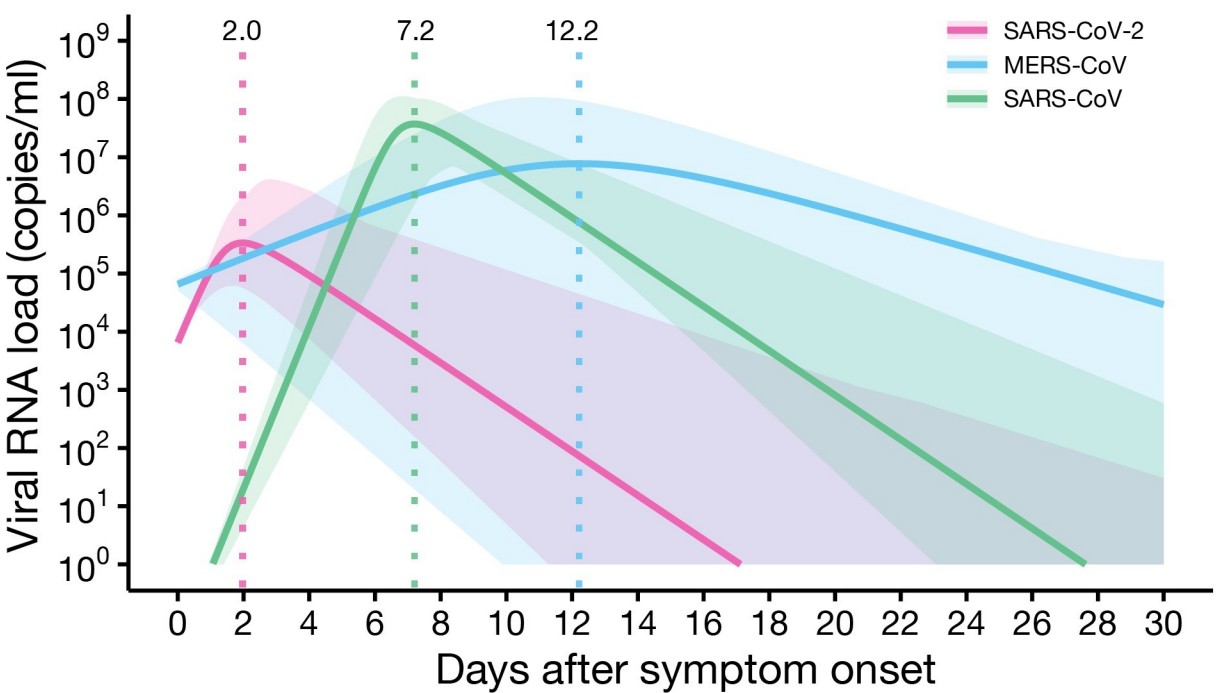

**Fig 1. Comparison of SARS-CoV-2, MERS-CoV, and SARS-CoV dynamics.** Expected viral load trajectories for SARS-CoV-2, MERS-CoV, and SARS-CoV infection are shown. The solid curves give the solution of Eqs 5 and 6 using estimated parameters (the best-fit population parameters), and the shaded regions correspond to 95% predictive intervals using the estimated parameters for each patient. The data underlying this figure are given in S1 Data.

### Evaluation of anti-SARS-CoV-2 therapies

Based on our mathematical model and estimated parameter values (Table 1), we conducted in silico experiments of possible anti-SARS-CoV-2 therapies to investigate the expected outcomes under hypothetical drug therapies (or vaccines) possessing different antiviral mechanisms of action (Fig 2). Specifically, drug efficacy (30% to 100%, i.e., $0.3 \leq \varepsilon, \eta, \theta \leq 1$) and timing of therapy initiation after symptom onset (i.e., $0 \leq t^* \leq 4$ days) were varied, and their influence on outcomes was investigated (see Methods) (Fig 2). We used reduction in the viral load area under the curve (AUC) and the fraction of target cells that remain uninfected 4 weeks after symptom onset as outcome measures. Without treatment, the AUC was $8.2 \times 10^5$ copies·day/ml, and almost no target cells remained after the course of infection (e.g., Figs 2 and 3).

**Blocking de novo infection.** One of the major mechanisms of action for antivirals is blocking de novo infection. Blocking can be induced by drugs including human neutralizing antibodies either in convalescent plasma or given as monoclonal antibodies, viral entry inhibitors, and/or antibodies raised by vaccination [5,29]. For example, a SARS-CoV-specific human monoclonal antibody bamlanivimab has received emergency use authorization by the US Food and Drug Administration for the treatment of SARS-CoV-2 [30].

Higher drug efficacy and earlier treatment initiation are associated with better outcomes: According to our model the AUC was reduced by 73%—and 74% of target cells remained uninfected after the course of infection—when treatment was initiated 1 day after symptom onset and the antiviral effectiveness was 90% (Fig 2). Very early treatment initiation is the key for better outcomes when using antiviral therapies.

According to our model, using a drug that blocks infection with 95% efficacy initiated 4 days after symptom onset, the AUC was reduced by only 14%, and only 2% of cells remain

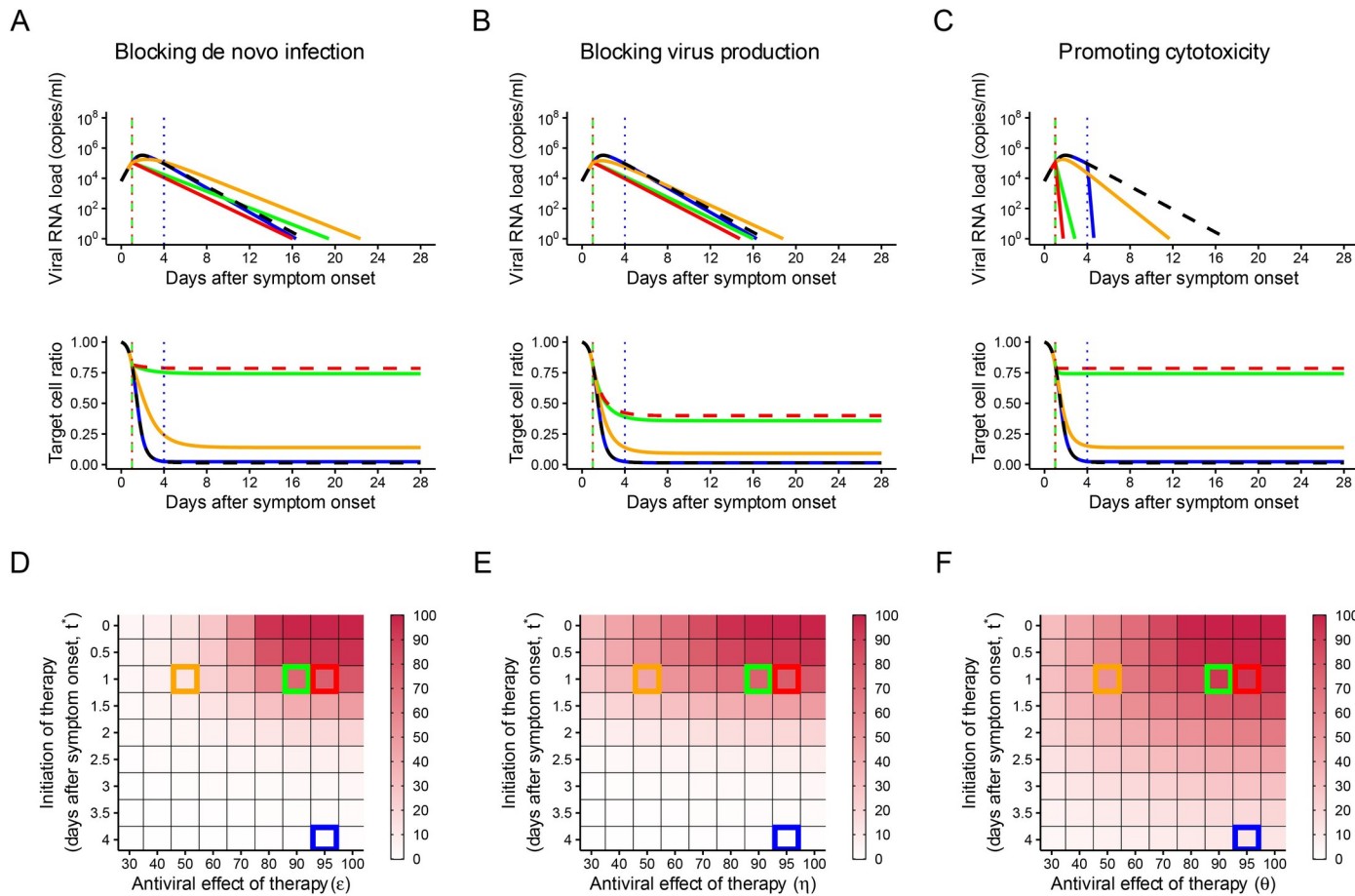

**Fig 2. Predicted outcomes under anti–SARS-CoV-2 monotherapies.** (A–C) Expected viral load and uninfected target cell proportion trajectories with and without treatment for the 3 different treatments. The black curves are without treatment. The blue curves are with treatment (efficacy is 95%) initiated at 4 days after symptom onset. The red, green, and orange curves are with treatment initiated at 1 day after symptom onset, but with different efficacy (95%, 90%, and 50%, respectively). The dotted vertical lines correspond to the timing of treatment initiation. (D–F) The heatmap shows the reduction in the viral load area under the curve (AUC) with treatment compared to without treatment. The timing of treatment initiation and treatment efficacy were varied. Darker colors indicate a larger reduction in the viral load AUC. The parameter setting used for the simulation in (A–C) is indicated by the same-colored squares in (D–F). The data underlying this figure are given in S2 Data.

uninfected (Fig 2A and 2D). This occurs because only a very small fraction of target cells remain uninfected after the viral load peak. After infection abates, target cells will replenish, but here we ignore this because we are evaluating the potential effects of therapy in preserving them. Note that viral shedding may last longer with treatment than without treatment if the antiviral efficacy is below 100% and initiated early. This is because substantial numbers of uninfected target cells remain at the time of treatment initiation, and the infection is driven by those uninfected cells but at a slower rate than without treatment.

We observed the same patterns for MERS-CoV and SARS-CoV (see S2A, S2D, S3A, and S3D Figs), except that treatment initiated a few days after symptom onset may be efficacious. As we observed in Fig 1, the viral load peak comes later for MERS-CoV and SARS-CoV than for SARS-CoV-2. Thus, even if treatment is initiated at 4 days after symptom onset (which is before viral load peak for those 2 viruses), improvement in the outcomes can be expected.

**Blocking virus production.** Most antiviral drugs inhibit intracellular virus replication. Lopinavir/ritonavir (HIV protease inhibitors), remdesivir (anti–Ebola virus disease candidate),

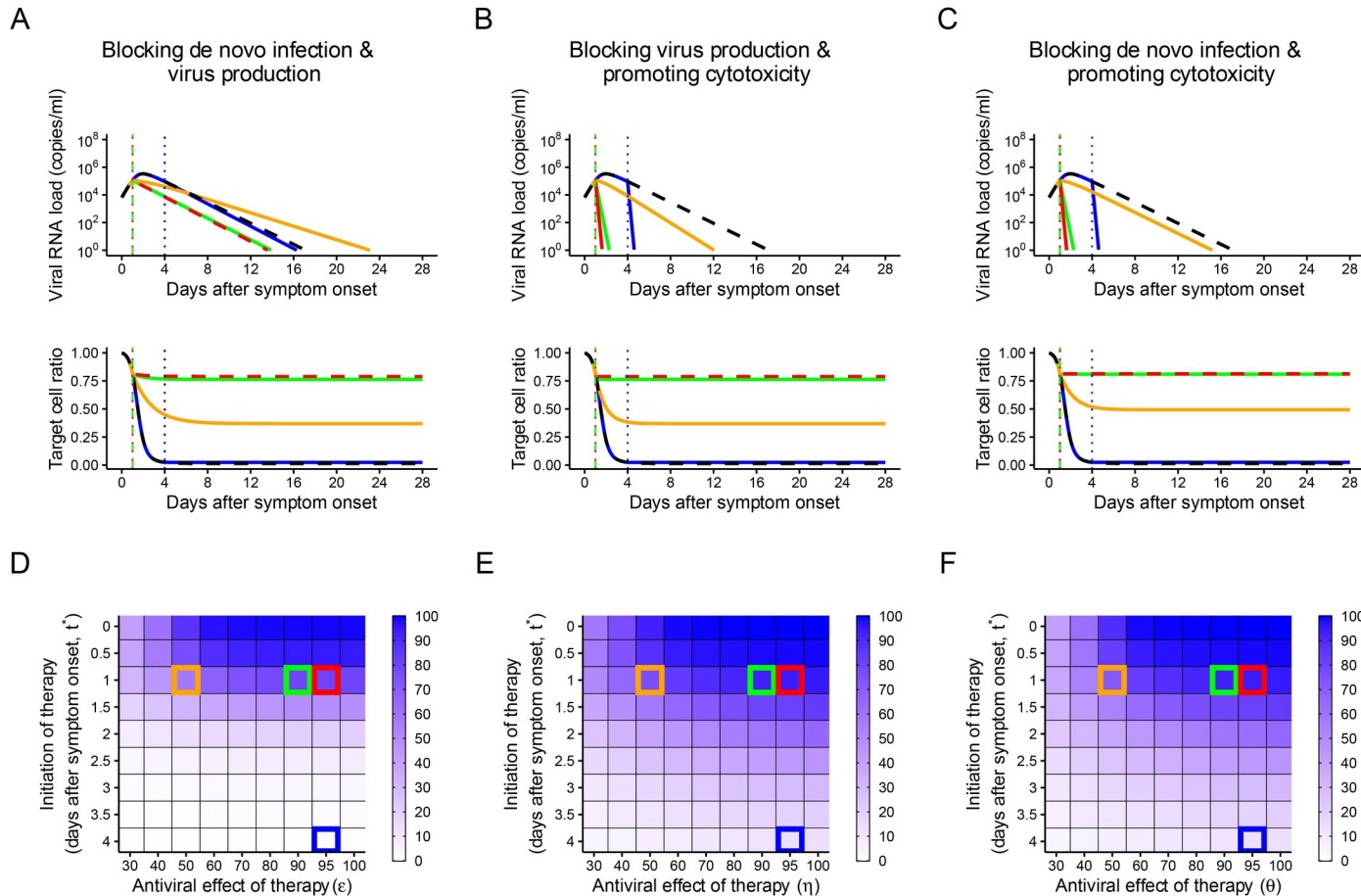

**Fig 3. Predicted outcomes under anti-SARS-CoV-2 combination therapies.** (A–C) Expected viral load and uninfected target cell proportion trajectories with and without treatment for the 3 combination therapies. We assumed the same efficacies and timing of treatment initiation for the 2 combined treatments. The black curves are without treatment. The blue curves are with treatment (efficacy is 95%) initiated at 4 days after symptom onset. The red, green, and orange curves are with treatment initiated at 1 day after symptom onset, but with different efficacy (95%, 90%, and 50%, respectively). The dotted vertical lines correspond to the time of treatment initiation. (D–F) The heatmap shows the reduction in the viral load area under the curve (AUC) with treatment compared to without treatment. The time of treatment initiation and the treatment efficacy were varied. Darker colors indicate a larger reduction in the viral load AUC. The parameter setting used for the simulation in (A–C) is indicated by the same-colored squares in (D–F). The data underlying this figure are given in S3 Data.

and other nucleoside analogues as well as interferon have the potential to suppress SARS-CoV-2 replication [31,32]. Similar to the findings for drugs blocking de novo infection, higher efficacy and earlier treatment are associated with better outcomes. According to our model the AUC was reduced by 76%, and 36% of the target cells remained uninfected after the course of infection, when treatment was initiated at 1 day after symptom onset and the antiviral effectiveness was 90% (Fig 2B and 2E).

In contrast, if treatment is started after the viral load peak, improvement in the outcomes cannot be expected even with a 100% inhibition rate. Similar patterns were observed for MERS-CoV and SARS-CoV (S2B, S2E, S3B, and S3E Figs). However, as 4 days after symptom onset is still before the viral load peak for these 2 viruses, substantial improvement in the outcomes is expected with treatment initiated 4 days after symptom onset for these 2 viruses (S2B, S2E, S3B, and S3E Figs).

**Promoting cytotoxicity.** Another possible antiviral mechanism is to promote cytotoxic effects. This could be done by stimulating adaptive immunity, including responses mediated by cytotoxic T lymphocytes and natural killer cells, by immunotherapy or vaccination, but the

effect would not be immediate. To be consistent with the other modes of drug action discussed above, in which we assume the drug takes effect immediately after administration, we envision a drug such as a viral-specific monoclonal antibody conjugated to a toxin, as used in cancer therapy [33], or a non-neutralizing viral-specific monoclonal antibody that could induce infected cell death by complement-mediated lysis or antibody-dependent cellular cytotoxicity. A neutralizing antibody with these effector functions could be considered the equivalent of combination therapy, which is discussed below. Compared with the other 2 therapeutic mechanisms of action (blocking de novo infection and virus production), the induction of cytotoxicity directly removes infected cells that produce viruses, and therefore it enhances the rate of viral load decay. After the viral peak, target cells are depleted and cytotoxicity-inducing therapy leads to noticeably more rapid declines in viral load (Fig 2C).

Thus, a 50% effective cytotoxicity-promoting antiviral, which by our definition (see Methods) causes the death rate of infected cells to double, initiated at day 1 after symptom onset results in an only slightly slower viral growth rate and an only slightly delayed time of viral load peak, but more rapid decay in viral load, than the other 2 therapeutic modes of action (blocking de novo infection and virus production) (Fig 2, orange curves). Moreover, cytotoxicity induction initiated after the viral load peak can still reduce the AUC. A 95% effective cytotoxicity-promoting antiviral initiated at 4 days after symptom onset reduces the AUC by 13%; however, only 2% of target cells remain uninfected because most of the target cells were already infected by the viral load peak (Fig 2C and 2F, blue curves). We confirmed that much later treatment initiation (13 days after symptom onset) with this type of antiviral still increases the rate of viral load decay (S4A Fig).

Overall, compared with the effects of the other 2 types of antivirals, the effect of promoting cytotoxicity on the AUC is less dependent on the magnitude of the antiviral effect and the timing of treatment initiation, although earlier treatment and more efficacy are positively associated with an increased reduction in the AUC.

We confirmed a similar pattern in the treatment effect on MERS-CoV and SARS-CoV infection (S2C, S2F, S3C, and S3F Figs). Given that their viral load peak comes later than that of SARS-CoV-2, treatment initiated at 4 days after symptom onset is predicted to still reduce the AUC and save uninfected target cells (see below).

To evaluate the effect of treatment promoting cytotoxicity initiated long after the viral load peak, we compared the effect of a 50% effective treatment initiated at 1 day and 13 days after symptom onset on all 3 coronaviruses (S4 Fig). The therapy initiated at 1 day delayed the time of the viral load peak particularly for MERS-CoV and SARS-CoV. When the treatment was initiated at 13 days, which is after the viral load peak, the viral load declined rapidly compared with treatment initiated at 1 day, because few target cells remained and thus new infection was limited.

The analysis of the treatment effect of drugs with 3 different modes of action revealed that the treatment strategy should be different for each type of drug. For example, using drugs that block de novo infection or virus production can avoid substantial target cell reduction if initiated before the viral load peak. Using a drug that promotes cytotoxicity is less time sensitive, and treatment initiated after the viral peak still can reduce the AUC. These findings suggest the possibility of a synergistic effect of combining drugs with different modes of action.

**Combination therapy.** In this section, we describe the effect of combining 2 different drug therapies from among the 3 described in the section above. In general, combinations of antiviral therapies are considered preferable when they synergistically enhance the antiviral effects, reduce the needed individual drug dose, and reduce the side effects compared with monotherapy [6,27,34–36]. Here, we focus on the synergistic antiviral effects on the model outcomes (i.e., reduction in the AUC and saving target cells from infection).

The 3 possible 2-drug combination therapies (i.e., blocking de novo infection and virus production, blocking virus production and promoting cytotoxicity, and blocking de novo infection and promoting cytotoxicity, shown in Fig 3A and 3D, 3B and 3E, and 3C and 3F, respectively) were simulated using the same assumptions as for the single drug therapies. All 3 combination therapies improved the antiviral effects compared to the corresponding monotherapies. As we expected, combining drugs with distinct modes of action, especially with a drug promoting cytotoxicity being one of them, more effectively reduced the AUC and saved target cells from infection. With monotherapy with a 50% antiviral effect initiated at 1 day after symptom onset, the AUC was reduced by 13%, 44%, and 54% with the drugs blocking de novo infection, blocking virus production, and promoting cytotoxicity, respectively (Fig 2D–2F), whereas the AUC was reduced by 58% or greater under combination therapy (Fig 3D–3F). Notably, combining a drug promoting cytotoxicity with 1 of the other 2 types of drugs compensated for the "weakness" of each treatment: No clear effect is expected from the drugs blocking de novo infection or virus production if initiated after the viral load peak.

From a biological point of view, promoting cytotoxicity is distinct from the other 2 mechanisms. Both blocking de novo infection and blocking virus production limit ongoing de novo infection, whereas promoting cytotoxicity enhances virus and infected cell removal independent of target cell availability. A broadly neutralizing antibody with potent effector functions that induced infected cell death would be a good therapeutic option as it induces 2 modes of action in 1 molecule. Antibodies of this type are being explored for HIV [37,38]. SARS-CoV-2 neutralizing antibodies are also in clinical development, and the role of their effector functions in providing protective activity is being examined [39]. Our analysis also implies that, if antiviral drugs induce immunomodulation as a bystander effect, even if the treatment is initiated after the viral load peak, they might be able to reduce viral load. We confirmed the same trends for MERS-CoV and SARS-CoV (S5B, S5E, S6B, and S6E Figs).

## Conclusions

To aid the development of antiviral drugs and treatment strategies for SARS-CoV-2 infection, we characterized the viral dynamics of SARS-CoV-2 and the related viruses SARS-CoV and MERS-CoV using a mathematical model. We further introduced the effect of antivirals with different modes of action in the model and explored the influence of the drug efficacy and timing of treatment initiation on the outcomes (viral load AUC and the fraction of target cells that remain uninfected). We found that $R_{S0}$ is larger for SARS-CoV-2 compared with MERS-CoV, and the difference in viral load peak timing was significantly different between SARS-CoV-2 and SARS-CoV. Some studies suggest that the viral load peak for SARS-CoV-2 occurs before the onset of symptoms [40,41], while other studies suggest that the viral load peak occurs within the first week of symptom onset [14,42–44]. Although it is difficult to accurately determine whether the peak is before or after symptom onset, since there are few viral load data available before the onset of symptoms, an earlier viral peak for SARS-CoV-2 is consistent with recent findings [14,40–44]. The larger $R_{S0}$ and earlier viral peak suggest it may be more difficult to treat SARS-CoV-2 infection than SARS-CoV and MERS-CoV with drug therapy that blocks viral production or de novo infection, because for these types of drugs, treatment initiation before the viral load peak is important to reduce viral load and save target cells from infection. The variations in parameter estimates among the individuals studied do not change our results on the importance of initiating antiviral therapy before the viral load peak (S7 Fig). The modeling of antivirals with different drug efficacies highlighted the importance of early initiation of treatments blocking de novo infection and virus production. In contrast, a treatment promoting cytotoxicity reduces AUC even when treatment is initiated after the viral load peak. Due to the uniqueness of the drugs promoting cytotoxicity

compared with the other 2 types of drugs, combination therapy with a drug promoting cytotoxicity and 1 of the 2 other drugs more effectively reduced the AUC and saved target cells from infection because the combination compensated for the weakness of each drug.

We used the viral load AUC and the fraction of target cells remaining uninfected as outcomes rather than the length of hospital stay, clinical improvement, severity, and mortality, which have been more commonly used as primary outcomes in clinical studies [45–52]. However, the outcomes should be determined on a case-by-case basis. For example, if the objective is to find or assess the effectiveness of a lifesaving treatment, then mortality should be used as a primary outcome. However, if the objective is to assess the effectiveness of antiviral treatment, the degree of viral load reduction might be a primary outcome. Indeed, viral-load-related outcomes have been used in multiple clinical studies for antivirals [15,46,53–57]. Further, viral load outcomes are particularly important for SARS-CoV-2 because many patients experience mild or no symptoms (i.e., asymptomatic cases) and yet are still isolated. To determine when to end isolation, it is frequently necessary to have a negative PCR test as well as disappearance of symptoms [14]. This is sensible as a strong association between viral load and infectiousness has been suggested [19].

Drug repurposing—reusing drugs already approved for specific purposes for other (new) purposes—is currently the major approach for rapidly deploying antiviral drugs for SARS-CoV-2. A number of drugs such as lopinavir and ritonavir [47,55]; chloroquine [48]; favipiravir [46]; interferon beta-1b, lopinavir/ritonavir, and ribavirin [58]; a nebulized form of interferon beta-1a [59]; and remdesivir [49,50] have been tested in clinical studies. However, the findings from such trials are not consistent: Some claim a significant effect, but the others do not, for the same drug. One of the major issues is poor study design [60]. Beyond that, we suspect that in some cases the treatment was not initiated early enough and may have yielded null findings even though the drug is effective, as we demonstrated in silico in this study of drugs blocking de novo infection and virus production. Indeed, the mean interval between symptom onset and hospitalization was 4.6 days during the COVID-19 epidemic in Shenzhen, China [28], which is longer than the interval between symptom onset and viral load peak for SARS-CoV-2 (2 days), suggesting that therapy is commonly started well after the viral load peak in hospitalized patients.

A limitation of our analysis is the simplicity of our mathematical model. However, this model is flexible and extendable. For example, we did not consider heterogeneity of target cells, and we assumed the death rate of an infected cell, δ, is constant. However, models with multiple types of target cells could be developed, and δ can be made time-dependent as was done in the case of HIV, where there were extensive viral load data [61,62]. Alternatively, equations can be introduced to explicitly model effector cell responses [63,64]. These approaches could be reflected in extended versions of our model if relevant data and supporting evidence become available. Indeed, several more complex models have been proposed to describe SARS-CoV-2 viral dynamics [17,18]. However, these complex mathematical models yielded similar conclusions as the simple model we employed about the need to initiate therapy early when using a typical antiviral that blocks viral production.

Development or identification of effective antiviral drugs is urgently needed. We believe our theoretical framework can at least partially explain why such drugs have not been identified (late treatment initiation) and could help design clinical studies and treatment strategies by assessing their potential effect on viral-load-related outcomes.

## Methods

### Study data

The longitudinal viral load data were extracted from clinical studies of SARS-CoV-2 [11–14], MERS-CoV [21,22], and SARS-CoV [23]. Only the data from individuals with 3 or more data

points above the detection limit were included in the analysis. The data from patients who received antiviral treatment during infection were excluded. We confirmed that ethics approval was obtained from the ethics committee at each institution, and that written informed consent was obtained from the patients or their next of kin in the original studies. The data were extracted from images in those publications using the program DataThief III (version 1.5, Bas Tummers; https://www.datathief.org). We converted cycle threshold (Ct) values reported in the above papers to viral RNA copy number values (copies/ml), where these quantities are inversely proportional to each other [65]. The following formula was used to convert Ct values ($y$) to viral RNA copies ($x$, in copies/ml): $\log_{10}(x) = ay+b$ with $a = -0.32$ and $b = 14.11$ [12]. S1 Table summarizes the data. The likelihood function accounted for censored data (i.e., data points under the detection limits) [66].

## Mathematical model

We used a simple target cell limited model to describe SARS-CoV-2, SARS-CoV, and MERS-CoV viral dynamics [20,24,67]. Target cell limited models have proved very valuable in understanding infection dynamics and therapy for chronic viral infections such as HIV [61,68], HCV [69], and HBV [70] and for acute infections such as influenza [71], West Nile virus [72], Zika virus [73], and SARS-CoV-2 [17,74,75]. Although the model does not explicitly describe immune responses, the effects of immune responses are implicitly included in model parameters such as the infection rate, which can be influenced by innate responses, and the death rate of infected cells, which can be influenced by adaptive immune responses. Because of the simplicity of the model, these parameters can be estimated and compared among the 3 different coronaviruses. The form of the model that we use was first introduced to model influenza infection [71] and is given by

$$\frac{dT(t)}{dt} = -\beta T(t)V(t), \tag{1}$$

$$\frac{dI(t)}{dt} = \beta T(t)V(t) - \delta I(t), \tag{2}$$

$$\frac{dV(t)}{dt} = pI(t) - cV(t), \tag{3}$$

where the variables $T(t)$, $I(t)$, and $V(t)$ are the number of uninfected target cells, the number of infected target cells, and the amount of virus at time $t$ (note: we used time after symptom onset as the timescale), respectively. Symptom onset is defined slightly differently between papers, but it essentially means when any coronavirus-related symptoms (fever, cough, and shortness of breath) appear [76]. The parameters $\beta$, $\delta$, $p$, and $c$ represent the rate constant for virus infection, the death rate of infected cells, the per cell viral production rate, and the per capita clearance rate of the virus, respectively. Since the clearance rate of the virus is typically much larger than the death rate of the infected cells in vivo [27,67,77], we made a quasi-steady state (QSS) assumption, $dV(t)/dt = 0$, and replaced Eq 3 with $V(t) = pI(t)/c$. Because data on the numbers of coronavirus RNA copies, $V(t)$, rather than the number of infected cells, $I(t)$, were available, $I(t) = cV(t)/p$ was substituted into Eq 2 to obtain

$$\frac{dV(t)}{dt} = \frac{p\beta}{c} T(t)V(t) - \delta V(t). \tag{4}$$

Furthermore, we replaced $T(t)$ by the fraction of target cells remaining at time $t$, i.e., $f(t) = T(t)/T(0)$, where $T(0)$ is the initial number of uninfected target cells. Note $f(0) = 1$. Accordingly, we

obtained the following simplified mathematical model, which we employed to analyze the viral load data in this study:

$$\frac{df(t)}{dt} = -\beta f(t)V(t),$$

(5)

$$\frac{dV(t)}{dt} = \gamma f(t)V(t) - \delta V(t),$$

(6)

where $\gamma = p\beta T(0)/c$ corresponds to the maximum viral replication rate under the assumption that target cells are continuously depleted during the course of infection. Thus, $f(t)$ is equal to or less than 1 and continuously declines.

In our analyses, the variable $V(t)$ corresponds to the viral load for SARS-CoV-2, MERS-CoV, and SARS-CoV (copies/ml). Because all of them cause acute infection, loss of target cells by physiological turnover can be ignored, considering the long lifespan of the target cells.

## The nonlinear mixed-effect model

Nonlinear mixed-effect modeling was used to fit the model to the longitudinal viral load data. The model includes both fixed effects (i.e., population parameters) and random effects. The random effects represent the difference among patients. The parameter value for patient $k$ is $\vartheta_k (= \vartheta \times e^{\pi_k})$, which is a product of a fixed effect, $\vartheta$, and a random effect, $e^{\pi_k}$. $\pi_k$ is assumed to follow the normal distribution: $N(0,\Omega)$. This approach allows us to estimate the parameters for patients with limited time point data, because the population parameters are estimated from not only their data, but all the patients' data. We used the viral type as a categorical covariate in estimating the parameters $\gamma$, $\beta$, and $V(0)$, which provide the lowest corrected Bayesian information criterion (BICc). Fixed effects and random effects were estimated using the stochastic approximation expectation-maximization algorithm and the empirical Bayes' method, respectively. The statistical differences of covariate for $\gamma$, $\beta$, and $V(0)$ were tested by the Wald test. Fitting was implemented using Monolix 2019R2 (https://www.lixoft.com) [78]. The estimated (fixed and individual) parameters and the initial values are listed in Table 1 and S2 Table. The viral load curve using the best-fit parameter estimates for each individual patient is shown, along with the data, in S1 Fig. Note that the mixed model approach has been used elsewhere in longitudinal viral load data analysis [17,73].

## In silico experiments for antiviral therapies

Based on the parameterized model for each virus, we investigated the antiviral effects of drugs with 3 different mechanisms of action—(i) blocking de novo infection, (ii) blocking virus production, and (iii) promoting cytotoxicity—on 2 outcomes: the reduction in the viral load AUC (i.e., $\int_0^{28} V(s)ds$) and the remaining fraction of target cells after the course of infection (i.e., $f(28)$). Note that we used 28 days after symptom onset as the upper bound for observation, because most viral load values are below the detection limit by this time and some previous clinical studies used health conditions (e.g., mortality) at 28 days (4 weeks) as a primary outcome [79]. In the simulation, the best-fit population parameters estimated by fitting the model to the data were used. We varied the time of treatment initiation after symptom onset, $t^*$, and the antiviral efficacy values $\varepsilon$, $\eta$, and $\theta$ to assess the dependency of them on the outcomes. Note that $t^* = 0$ corresponds to therapy initiated immediately after symptom onset.

We modeled viral load dynamics under antiviral treatment with the 3 different mechanisms of action as follows.

**Blocking de novo infection.** The viral dynamics under antiviral treatment for blocking de novo infection is modeled as follows:

$$\frac{df(t)}{dt} = -(1 - \varepsilon H(t))\beta f(t) V(t), \tag{7}$$

$$\frac{dV(t)}{dt} = (1 - \varepsilon H(t))\gamma f(t) V(t) - \delta V(t), \tag{8}$$

where $H(t)$ is the Heaviside step function defined as $H(t) = 0$ if $t < t^*$; otherwise, $H(t) = 1$. $t^*$ is the time of treatment initiation, and $\varepsilon$ is the treatment efficacy: $0 < \varepsilon \leq 1$. $\varepsilon = 1$ implies de novo infection is 100% inhibited.

**Blocking virus production.** The virus dynamics under treatment for blocking virus production is modeled as follows:

$$\frac{dV(t)}{dt} = (1 - \eta H(t))\gamma f(t) V(t) - \delta V(t), \tag{9}$$

where $\eta$ is the treatment efficacy: $0 < \eta \leq 1$. $\eta = 1$ indicates that virus production from infected cells is fully inhibited. Note that the difference between blocking de novo infection and virus production is that the drugs in the former model reduce $\beta$, whereas the drugs in this model reduce $p$, in the full model, i.e., Eqs 1–3.

**Promoting cytotoxicity.** The virus dynamics under the antiviral treatment of promoting cytotoxicity (or increasing the death rate of infected cells) is modeled as follows:

$$\frac{dV(t)}{dt} = \gamma f(t) V(t) - \left(\frac{1}{1 - \theta H(t)}\right)\delta V(t), \tag{10}$$

where $\theta$ is the treatment efficacy: $0 < \theta \leq 1$. $\theta = 1$ indicates that the drug is 100% effective and causes the immediate death of an infected cell. No drug is expected to be 100% effective. A 50% effective drug would cause a 2-fold increase in the death rate, and a 90% effective drug would cause a 10-fold increase.

**Combination therapy.** The virus dynamics under therapies combining the 3 types of drugs is modeled as follows:

$$\frac{df(t)}{dt} = -(1 - \varepsilon H(t))\beta f(t) V(t), \tag{11}$$

$$\frac{dV(t)}{dt} = (1 - \varepsilon H(t))(1 - \eta H(t))\gamma f(t) V(t) - \left(\frac{1}{1 - \theta H(t)}\right)\delta V(t). \tag{12}$$

In the simulation, we assumed any 2 therapies are combined (thus 1 of the 3 parameters is set as 0).

## Computation of $L$, $R_{S0}$, $C^*$, and $T_p$ and statistical test for the difference between viruses

Based on the estimated parameters, we calculated several quantities for each virus: the duration of virus production ($L = 1/\delta$), the reproduction number ($R_{S0} = \gamma/\delta$) at symptom onset, and the critical inhibition level ($C^* = 1 - 1/R_{S0}$). Further, the time from symptom onset to the viral load peak ($T_p$) was calculated by running the model using estimated (fixed and individual) parameters and the initial values. The difference in $T_p$ was tested by the jackknife test [80,81]. To evaluate statistical differences for $R_{S0}$ and $C^*$, we applied the Wald test as well.

## Supporting information

**S1 Data. Original numerical values for Fig 1.**
(XLSX)

**S2 Data. Original numerical values for Fig 2.**
(XLSX)

**S3 Data. Original numerical values for Fig 3.**
(XLSX)

**S4 Data. Original numerical values for S1 Fig.**
(XLSX)

**S5 Data. Original numerical values for S2 Fig.**
(XLSX)

**S6 Data. Original numerical values for S3 Fig.**
(XLSX)

**S7 Data. Original numerical values for S4 Fig.**
(XLSX)

**S8 Data. Original numerical values for S5 Fig.**
(XLSX)

**S9 Data. Original numerical values for S6 Fig.**
(XLSX)

**S10 Data. Original numerical values for S7 Fig.**
(XLSX)

**S1 Fig. Viral load trajectory for individual patients with SARS-CoV-2, MERS-CoV, and SARS-CoV infection.** (A–C) The estimated viral load for each individual patient (solid lines) along with the observed data (closed dots) are depicted using the best-fit parameter estimates. The dotted horizontal lines are the detection limits. Note that the detection limits for SARS-CoV-2 were 68 (Singapore, Korea), 15 (China), and 33.3 (Germany) copies/ml. The detection limits for MERS-CoV were 1,000 (Korea) and 15 (Saudi Arabia) copies/ml. The red closed dots at the detection limit represent the data points where the virus was detected but below the detection limit. The data underlying this figure are given in S4 Data.
(DOCX)

**S2 Fig. Predicted outcomes under anti-MERS-CoV monotherapies.** (A–C) Expected viral load and uninfected target cell proportion trajectories with and without treatment for the 3 different treatments. The black curves are without treatment. The blue curves are with treatment (efficacy is 95%) initiated at 16 days after symptom onset. The red and green curves are with treatment initiated at 4 days after symptom onset, but with different efficacy (95% and 90%). The dotted vertical lines correspond to the timing of treatment initiation. (D–F) The heatmap shows the reduction in the viral load AUC with treatment compared to without treatment. The timing of treatment initiation and treatment efficacy were varied. Darker colors indicate a larger reduction in the viral load AUC. The parameter setting used for the simulation in (A–C) is indicated by the same color in (D–F). The data underlying this figure are given in S5 Data.
(TIF)

**S3 Fig. Predicted outcomes under anti-SARS-CoV monotherapies.** (A–C) Expected viral load and uninfected target cell proportion trajectories with and without treatment for the 3

different treatments. The black curves are without treatment. The blue curves are with treatment (efficacy is 95%) initiated at 16 days after symptom onset. The red and green curves are with treatment initiated at 4 days after symptom onset, but with different efficacy (95% and 90%). The dotted vertical lines correspond to the timing of treatment initiation. (D–F) The heatmap shows the reduction in the viral load AUC with treatment compared to without treatment. The timing of treatment initiation and treatment efficacy were varied. Darker colors indicate a larger reduction in the viral load AUC. The parameter setting used for the simulation in (A–C) is indicated by the same color in (D–F). The data underlying this figure are given in S6 Data.
(TIF)

**S4 Fig. Predicted outcomes under drugs promoting cytotoxicity initiated at 1 and 13 days after symptom onset.** The dotted black curve corresponds to the viral load (top) and uninfected target cell proportion (bottom) trajectories without treatment promoting cytotoxicity for the 3 coronaviruses. The blue and green curves are with treatment with 50% efficacy initiated at 1 or 13 days after symptom onset. The data underlying this figure are given in S7 Data.
(TIF)

**S5 Fig. Predicted outcomes under anti-MERS-CoV combination therapies.** (A–C) Expected viral load and uninfected target cell proportion trajectories with and without treatment for the 3 combination therapies. We assumed the same efficacies and timing of treatment initiation for the 2 combined treatments. The black curves are without treatment. The blue curves are with treatment (efficacy is 95%) initiated at 16 days after symptom onset. The red and green curves are with treatment initiated at 4 days after symptom onset, but with different efficacy (95% and 90%). The dotted vertical lines correspond to the timing of treatment initiation. (D–F) The heatmap shows the reduction in the viral load AUC with treatment compared to without treatment. The timing of treatment initiation and treatment efficacy were varied. Darker colors indicate a larger reduction in the viral load AUC. The parameter setting used for the simulation in (A–C) is indicated by the same color in (D–F). The data underlying this figure are given in S8 Data.
(TIF)

**S6 Fig. Predicted outcomes under anti-SARS-CoV combination therapies.** (A–C) Expected viral load and uninfected target cell proportion trajectories with and without treatment for the 3 combination therapies. We assumed the same efficacies and timing of treatment initiation for the 2 combined treatments. The black curves are without treatment. The blue curves are with treatment (efficacy is 95%) initiated at 16 days after symptom onset. The red and green curves are with treatment initiated at 4 days after symptom onset, but with different efficacy (95% and 90%). The dotted vertical lines correspond to the timing of treatment initiation. (D–F) The heatmap shows the reduction in the viral load AUC with treatment compared to without treatment. The timing of treatment initiation and treatment efficacy were varied. Darker colors indicate a larger reduction in the viral load AUC. The parameter setting used for the simulation in (A–C) is indicated by the same color in (D–F). The data underlying this figure are given in S9 Data.
(TIF)

**S7 Fig. Predicted outcomes under anti-SARS-CoV-2 monotherapies considering interindividual variation in parameters.** Expected viral load and uninfected target cell proportion trajectories with and without treatment for the 3 different treatments. The black curves are without treatment. The blue and red curves are with treatment (efficacy is 95%) initiated at 1 day and at 4 days after symptom onset, respectively. The shaded regions correspond to 95%

predictive intervals. The data underlying this figure are given in S10 Data.
(TIF)

**S1 Table. Summary of the data used in our analysis.**
(DOCX)

**S2 Table. Estimated parameters and initial values for each patient.**
(DOCX)

**S1 Text. Study data.**
(DOCX)

## Author Contributions

**Conceptualization:** Keisuke Ejima, Yasuhisa Fujita, Shingo Iwami.

**Data curation:** Kwang Su Kim.

**Formal analysis:** Kwang Su Kim, Shoya Iwanami, Shinji Nakaoka, Robin N. Thompson, Shingo Iwami.

**Funding acquisition:** Shingo Iwami.

**Investigation:** Kwang Su Kim, Keisuke Ejima, Shoya Iwanami, Yasuhisa Fujita, Hirofumi Ohashi, Yoshiki Koizumi, Yusuke Asai, Shinji Nakaoka, Koichi Watashi, Kazuyuki Aihara, Robin N. Thompson, Ruian Ke, Alan S. Perelson, Shingo Iwami.

**Methodology:** Kwang Su Kim, Keisuke Ejima, Shoya Iwanami, Shinji Nakaoka, Robin N. Thompson, Ruian Ke, Alan S. Perelson, Shingo Iwami.

**Project administration:** Keisuke Ejima, Alan S. Perelson, Shingo Iwami.

**Supervision:** Keisuke Ejima, Alan S. Perelson, Shingo Iwami.

**Validation:** Keisuke Ejima, Ruian Ke, Shingo Iwami.

**Visualization:** Shoya Iwanami, Shingo Iwami.

**Writing – original draft:** Kwang Su Kim, Keisuke Ejima, Shoya Iwanami, Yasuhisa Fujita, Hirofumi Ohashi, Yoshiki Koizumi, Yusuke Asai, Shinji Nakaoka, Koichi Watashi, Kazuyuki Aihara, Robin N. Thompson, Ruian Ke, Alan S. Perelson, Shingo Iwami.

**Writing – review & editing:** Kwang Su Kim, Keisuke Ejima, Shoya Iwanami, Yoshiki Koizumi, Kazuyuki Aihara, Ruian Ke, Alan S. Perelson, Shingo Iwami.

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
