## [Editor Report · Decision Letter 0]

18 Dec 2020

Dear Dr Iwami, 

Thank you for submitting this revised version of your manuscript entitled "Quantitative comparison of SARS-CoV-2, MERS-CoV and SARS-CoV dynamics: Implications for therapy" for consideration as a Research Article by PLOS Biology.

Your revisions have now been evaluated by the PLOS Biology editorial staff, and by the Academic Editor, and I'm writing to let you know that we would like to send your submission out for re-review.

Please re-submit your manuscript within two working days, i.e. by Dec 22 2020 11:59PM.

Kind regards,

Roli Roberts

Senior Editor

PLOS Biology

---

## [Decision Letter · Decision Letter 1]

23 Jan 2021

Dear Shingo,

Thank you for submitting your revised Research Article entitled "Quantitative comparison of SARS-CoV-2, MERS-CoV and SARS-CoV dynamics: Implications for therapy" for publication in PLOS Biology. I've now obtained advice from two of the original reviewers and have discussed their comments with the Academic Editor. Please note that reviewer #3 is the person who had told us that they had reviewed your study for another journal, but to whose comments you responded. Reviewer #2 is the original PLOS Biology reviewer #2. Reviewer #1 was not able to re-review.

Based on the reviews, we will probably accept this manuscript for publication, assuming that you will modify the manuscript to address the remaining points raised by the reviewers. Please also make sure to address the data and other policy-related requests noted at the end of this email.

IMPORTANT:

a) Please attend to the remaining requests from reviewer #3.

b) Please update your co-author's funding information (as discussed in our email conversation).

c) Please change your title to something that makes it clear that this is a modelling study, and that reduces the emphasis on implications for therapy. We suggest the following: "A quantitative model to compare within-host SARS-CoV-2, MERS-CoV and SARS-CoV dynamics provides insights into the pathogenesis of SARS-CoV-2"

d) Please address my Data Policy requests (see further down) by supplying the data and/or code that underlies the Figures.

We expect to receive your revised manuscript within two weeks. Your revisions should address the specific points made by each reviewer. 

-  a cover letter that should detail your responses to any editorial requests, if applicable

*Published Peer Review History*

*Early Version*

Best wishes,

Roli

Senior Editor,

rroberts@plos.org,

PLOS Biology

DATA POLICY:

Regardless of the method selected, please ensure that you provide the individual numerical values that underlie the summary data displayed in the following figure panels as they are essential for readers to assess your analysis and to reproduce it: data for Fig S1, and data or code needed to generate Figs 1, 2ABCDEF, 3ABCDEF, S2ABCDEF, S3ABCDEF, S4ABC, S5ABCDEF, S6ABCDEF, S7. NOTE: the numerical data provided should include all replicates AND the way in which the plotted mean and errors were derived (it should not present only the mean/average values).

REVIEWERS' COMMENTS:

Reviewer #2:

The revised paper with much more patient data does help to address my concerns. Also, the more detailed explanations of the roles of different drug types now make sense to me. I have no further comments.

Reviewer #3:

I thank the authors for their thorough responses to my previous comments. I think the manuscript is more focused and more clearly states what the modelling can show. 

I just have two remaining comments:

1. For the target cell models it is mentioned that the parameters implicitly include the immunity. Do these parameters (delta for example as mentioned in the discussion) change over time. If not, they are missing an important thing that immunity does to control infection which is proliferate over time. Could the authors please comment on this. 

2. I would still like to see a little more discussion in line 306-313 on the endpoints used in trials that are generally not viral, and how what it is that is important for the drugs to be able to do (I would think reduce severity and death) and how this is not what is discussed in this paper.

---

## [Editor Report · Decision Letter 2]

1 Feb 2021

Dear Shingo,

On behalf of my colleagues and the Academic Editor, Bill Sugden, I'm pleased to say that we can in principle offer to publish your Research Article "A quantitative model used to compare within-host SARS-CoV-2, MERS-CoV and SARS-CoV dynamics provides insights into the pathogenesis and treatment of SARS-CoV-2" in PLOS Biology, provided you address any remaining formatting and reporting issues. These will be detailed in an email that will follow this letter and that you will usually receive within 2-3 business days, during which time no action is required from you. Please note that we will not be able to formally accept your manuscript and schedule it for publication until you have made the required changes.

PRESS: We frequently collaborate with press offices. If your institution or institutions have a press office, please notify them about your upcoming paper at this point, to enable them to help maximise its impact. If the press office is planning to promote your findings, we would be grateful if they could coordinate with biologypress@plos.org. If you have not yet opted out of the early version process, we ask that you notify us immediately of any press plans so that we may do so on your behalf.

Thank you again for supporting Open Access publishing. We look forward to publishing your paper in PLOS Biology. 

Sincerely,

Roli

Roland G Roberts, PhD 

Senior Editor 

PLOS Biology